# Quantitative Analysis of Daporinad (FK866) and Its In Vitro and In Vivo Metabolite Identification Using Liquid Chromatography-Quadrupole-Time-of-Flight Mass Spectrometry

**DOI:** 10.3390/molecules27062011

**Published:** 2022-03-21

**Authors:** Minjae Park, Byeong Ill Lee, Jangmi Choi, Yuri Park, Seo-Jin Park, Jeong-Hyeon Lim, Jiyu Lee, Young G. Shin

**Affiliations:** Institute of Drug Research and Development, College of Pharmacy, Chungnam National University, 99 Daehak-ro, Yuseong-gu, Daejeon 34134, Korea; minjae.park.cnu@gmail.com (M.P.); byungill.lee.cnu@gmail.com (B.I.L.); jangmi.choi.cnu@gmail.com (J.C.); yuri.park.cnu@gmail.com (Y.P.); seojin.park.cnu@gmail.com (S.-J.P.); jeonghyeon.lim.cnu@gmail.com (J.-H.L.); jiyu.lee.cnu@gmail.com (J.L.)

**Keywords:** Daporinad (FK866), NAMPT inhibitor, LC-qTOF-MS, qualification, pharmacokinetic, metabolism

## Abstract

Daporinad (FK866) is one of the highly specific inhibitors of nicotinamide phosphoribosyl transferase (NAMPT) and known to have its unique mechanism of action that induces the tumor cell apoptosis. In this study, a simple and sensitive liquid chromatography–quadrupole-time-of-flight–mass spectrometric (LC-qTOF-MS) assay has been developed for the evaluation of drug metabolism and pharmacokinetics (DMPK) properties of Daporinad in mice. A simple protein precipitation method using acetonitrile (ACN) was used for the sample preparation and the pre-treated samples were separated by a C18 column. The calibration curve was evaluated in the range of 1.02~2220 ng/mL and the quadratic regression (weighted 1/concentration^2^) was used for the best fit of the curve with a correlation coefficient ≥ 0.99. The qualification run met the acceptance criteria of ±25% accuracy and precision values for QC samples. The dilution integrity was verified for 5, 10 and 30-fold dilution and the accuracy and precision of the dilution QC samples were also satisfactory within ±25% of the nominal values. The stability results indicated that Daporinad was stable for the following conditions: short-term (4 h), long-term (2 weeks), freeze/thaw (three cycles). This qualified method was successfully applied to intravenous (IV) pharmacokinetic (PK) studies of Daporinad in mice at doses of 5, 10 and 30 mg/kg. As a result, it showed a linear PK tendency in the dose range from 5 to 10 mg/kg, but a non-linear PK tendency in the dose of 30 mg/kg. In addition, in vitro and in vivo metabolite identification (Met ID) studies were conducted to understand the PK properties of Daporinad and the results showed that a total of 25 metabolites were identified as ten different types of metabolism in our experimental conditions. In conclusion, the LC-qTOF-MS assay was successfully developed for the quantification of Daporinad in mouse plasma as well as for its in vitro and in vivo metabolite identification.

## 1. Introduction

In recent years, many studies have been conducted on the regulation of cancer metabolism as new therapeutic targets [1,2,3,4,5]. This is because cancer metabolism-related agents can potentially enhance the therapeutic effect when combined with conventional chemotherapies [6,7]. Recently, the biochemical pathway of nicotinamide adenine dinucleotide (NAD) has received much attention among these cancer metabolism researches [8,9,10,11,12,13,14]. The NAD is a major co-enzyme of glyceraldehyde-3-phosphate dehydrogenase (G3PDH) in glycolysis, a major glucose metabolic pathway in cancer cells, and is also involved in tricarboxylic acid (TCA) cycle and oxidative phosphorylation in cancer cells [15,16]. There are also evidences that some of chemotherapy-resistant tumors might rely on NAD-dependent biochemical pathways based on the high levels of NAD observed in the tumors [17].

There are three types of biochemical pathways in NAD such as de novo, preiss-handler, and salvage pathways. Among them, the salvage pathway consists of two-step pathway in which NAD is first synthesized from nicotinamide (NAM) to nicotinamide mononucleotide (NMN) by nicotinamide phosphoribosyltransferase (NAMPT) enzyme followed by NMN to NAD by nicotinamide mononucleotide adenylyltransferase (NMNAT) enzyme [8,9,10,11]. In this two-step pathway, the reaction of NAMPT is mainly the rate-limiting step, and the high levels of NAD are observed in many chemotherapy-resistant cancer cells due to over-expression of the corresponding enzyme [8,10,18,19]. Therefore, it was suggested that inhibition of NAMPT would be able to reduce the NAD levels, and, therefore, could prevent the growth of tumors.

Daporinad (FK866) is an anti-cancer agent recently being developed as one of the highly selective, non-competitive NAMPT inhibitors and its structure is shown in Figure 1. Daporinad has demonstrated excellent tumor regression efficacy in various in vitro and in vivo experiments [20,21,22,23,24,25,26,27,28,29]. Daporinad completed the phase 2 clinical trial [29,30], and after that, efficacy studies on various tumor targets are ongoing [31,32].

In this study, we developed a simple and sensitive liquid chromatography-quadrupole time of flight mass spectrometric (LC-qTOF-MS) assay in order to explore the pharmacokinetics (PK) properties of Daporinad in mouse plasma using fit-for-purpose bioanalytical method qualification criteria. A simple protein precipitation with acetonitrile (ACN) followed by centrifugation was used for the PK samples as well as the MetID samples. Recently Qu and Liu published a paper regarding the MetID of Daporinad [33]. The approach carried out by Qu and Liu using high-resolution mass spectrometer sounds reasonable from the analytical strategy perspectives but the coverage of MetID was limited due to their incubation assay such as in vitro liver microsomes with NADPH which can reflect phase I metabolism only. In addition, in vivo MetID was not explored in their paper, which is more critical than the in vitro MetID to understand the relationship of in vivo PK and in vivo metabolism. To the author’s best knowledge, our manuscript is the first study to evaluate the PK properties in mouse plasma as well as the MetID of Daporinad, both in vitro and in vivo.

## 2. Results and Discussion

### 2.1. LC-qTOF-MS Method Development and Qualification for Daporinad in Mouse Plasma

#### 2.1.1. Method Development for Quantification of Daporinad in Mouse Plasma

A method for the quantification of Daporinad in mouse plasma was developed using an LC-qTOF-MS assay. Calibration curve consists of duplicate standard (STD) samples at eight levels of concentration and quality control (QC) samples at three levels. The calibration curve was established in the range of 1.02~2220 ng/mL by the quadratic regression (weighted 1/concentration^2^) with a correlation coefficient ≥0.99. The calibration curve and the chromatogram of the lower limit of quantification (LLOQ) level for Daporinad in mouse plasma are shown in Figure 2.

Assay performance was determined by assessing the mean accuracy and precision of QC samples with three levels of concentrations (15.0, 165, and 1820 ng/mL). The intra- and inter-run accuracy and precision were evaluated in triplicates and the results are shown in Table 1. As a result, the qualification run met the in-house acceptance criteria of ±25% of the nominal values from the coefficient of variation (CV) and bias perspectives for discovery non-GLP study which are also a well-known criteria for the discovery stage studies in pharmaceutical industry [34]. The dilution integrity was evaluated by preparing three different concentrations of dilution QC samples at 5-, 10- and 30-fold dilutions, respectively. The result for dilution integrity is presented in Table 2.

#### 2.1.2. Preliminary Stability Tests for Daporinad in Mouse Plasma

The preliminary stability tests showed that Daporinad in mouse plasma was stable for at least 4 h at room temperature (RT) which is sufficient for the time period expected for routine sample preparation. Daporinad was stable in mouse plasma for −80 °C three cycles of freeze–thawing and also stable during at least 2 weeks at −80 °C storage condition. Each stability results are shown in Table 3.

### 2.2. Application for Intravenous (IV) Pharmacokinetic (PK) Studies in Mouse

The developed LC-qTOF-MS method was successfully applied to obtain the PK parameters of Daporinad in mouse plasma following IV administration at the doses of 5, 10 and 30 mg/kg. The concentrations of most PK samples were within the qualified calibration curve except some PK samples at early time point samples. Therefore, early time point PK samples were diluted appropriately with blank mouse plasma to ensure that every sample concentration was within the calibration curve range. The time to concentration profile of Daporinad is shown in Figure 3. The PK parameters were calculated with the non-compartmental analysis (NCA) using WinNonlin (version 8.0.0) and the results are summarized in Table 4.

The PK results show that the maximum concentration (C_max_) and the area under the curve (AUC) for Daporinad were dose proportional in the dose range from 5 to 10 mg/kg in mouse plasma. However, in the dose range from 10 to 30 mg/kg, the C_max_ was dose-proportional, but the AUC appeared to increase supra-proportional, possibly due to the saturation of the elimination pathway including either metabolic enzymes or transporters in this dose range.

### 2.3. In Vitro and In Vivo Metabolite Identification (Met ID) for Daporinad

The metabolites of Daporinad were investigated using the LC-qTOF-MS assay to understand the clearance (Cl) change of Daporinad in the previous PK study. The fragmentation pattern and the retention time of Daporinad were used as a reference to elucidate the structures of proposed metabolites. The extracted ion chromatograms for the parent and the proposed metabolites of Daporinad from in vitro and in vivo Met ID experiments are present in Figure 4, and the metabolic pathways of Daporinad are also shown in Figure 5. Under the current in vitro and in vivo experimental conditions, twenty-five metabolites were identified, as shown in Table 5. These metabolites were characterized by amide hydrolysis (M1), di-saturation (M3 and M9), mono-oxidation (M4, M8, M17, M22 and M24), di-oxidation (M5, M6, M7, M10 and M13), tri-oxidation (M11 and M15), saturation (M12), desaturation (M16, M23 and M25), desaturation + oxidation (M2, M14, M18 and M19), amide hydrolysis followed by carboxylation (M20), and amide hydrolysis followed by hydroxylation (M21). The results suggested that ten different metabolic pathway，of them were phase Ⅰ metabolites. All metabolites identified from in vivo mouse plasma appeared to cover the metabolites generated by mouse and human liver microsomes.

As a part of further investigation, a post-preparation analysis using titanium chloride solution (TiCl_3_) was also performed to evaluate whether N-oxide metabolites on the pyridine group of Daporinad were presented. In general, TiCl_3_ is known to covert N-oxide metabolite back to the parent drug [35,36]. As a result, six metabolites (M6, M10, M11, M18, M22 and M25) were identified to have an N-oxide moiety in their structures as shown in Figure 6. Through this additional post-preparation analysis, more detailed structural elucidation of metabolites was performed.

Daporinad

The *m/z* of the [M+H]^+^ form of Daporinad was 392.23. The product ion scan of *m/z* 392.23 led to the formation of fragment ions at *m/z* 261.20, 244.17, 140.14, 132.04, 105.03 and 104.05. The fragment ions at *m/z* 261.20 and 132.04 were likely formed by breaking the amide bond (*m/z* 392.23). Two other ions at *m/z* 244.17 and 104.05 were likely generated via the neutral loss of ammonia (NH_3_) and carbon monoxide (CO) from the corresponding ions at *m/z* 261.20 and 132.04, respectively. The fragment ion at *m/z* 244.17 also confirmed two product ions at *m/z* 105.03 and *m/z* 140.14 by accurate mass measurement. The detailed fragmentation patterns of Daporinad are shown in Figure 7a.

M1

The *m/z* of the [M+H]^+^ form of M1 was 261.20, which means that the M1 has 131 amu less than Daporinad. The product ion scan of *m/z* 261.20 led to the formation of fragment ions at *m/z* 140.14 and 105.03. Two unchanged fragment ions indicated that the metabolism of M1 was occurred in the amide bond of parent drug. These data suggested that M1 was formed via the cleavage of the amide bond of Daporinad. The detailed fragmentation patterns of M1 are shown in Figure 7b.

M2 and M14

The *m/z* of the [M+H]^+^ form of M2 and M14 was 406.21. M2 and M14 have 14 amu less than Daporinad which suggests that desaturation and oxidation were likely occurred. The product ion scan of *m/z* 406.21 led to the formation of fragment ions at *m/z* 275.18, 258.15, 154.12, 132.04, 105.03 and 104.05. In the fragment pattern of M2 and M14, three fragment ions (275.18, 258.15 and 154.12) increased by 14 amu suggested that metabolism of M2 and M4 were occurred in the 4-butyl piperidine of Daporinad. These results suggest that M2 and M14 metabolites are the oxidation and desaturation metabolites on the 4-butyl piperidine moiety of Daporinad. The detailed fragmentation patterns of M2 and M14 are shown in Figure 7c.

M3 and M9

The *m/z* of the [M+H]^+^ form of M3 and M9 was 388.20. M3 and M9 have 4 amu less than Daporinad which suggests di-desaturation. The product ion scan of *m/z* 388.20 led to the formation of fragment ions at *m/z* 257.16, 240.14, 136.11, 132.04, 105.03 and 104.05. Among these fragments, three fragment ions (257.16, 240.14 and 136.11) decreased by 4 amu indicated that the metabolism of M3 and M9 was occurred in 4-butyl piperidine of Daporinad. These results suggest that M3 and M9 were di-desaturation metabolites on the 4-butyl piperidine moiety of Daporinad. The detailed fragmentation patterns of M3 and M9 are shown in Figure 7d.

M4, M8 and M17

The *m/z* of the [M+H]^+^ form of M4, M8 and M17 was 408.23. M4, M8 and M17 have 16 amu more than Daporinad suggesting mono-oxidation metabolites. The product ion scan of *m/z* 408.23 led to the formation of fragment ions at *m/z* 277.19, 260.16, 156.14, 132.04, 105.03 and 104.05. Among them, three fragment ions (277.19, 260.19 and 156.14) increased by 16 amu indicated that the metabolism of M4, M8 and M17 were occurred in 4-butyl piperidine of parent drug. The detailed fragmentation patterns of M4, M8 and M17 are shown in Figure 7e.

M5

The *m/z* of the [M+H]^+^ form of M5 was 424.22. M5 has 32 amu more than Daporinad, suggesting the di-oxidation metabolite. The product ion scan of *m/z* 424.22 led to the formation of fragment ions at *m/z* 293.19, 276.16, 172.13, 132.04, 105.03 and 104.05. The detailed fragmentation patterns of M5 are shown in Figure 7f.

M6, M7, M10 and M13

The *m/z* of the [M+H^]+^ form of M6, M7, M10 and M13 was also 424.22, suggesting di-oxidation metabolites. The product ion scan of *m/z* 424.22 led to the formation of fragment ions at *m/z* 277.19, 156.14, 148.04, 120.04 and 105.03. Particularly, four unique fragment ions (277.19, 156.14, 148.04 and 120.04) increased by 16 amu showed that the metabolism of M6, M7, M10 and M13 were likely occurred in 4-butyl piperidine and pyridine-acryloyl moiety of parent drug, respectively. The detailed fragmentation patterns of M6, M7, M10 and M13 are shown in Figure 7g.

M11 and M15

The *m/z* of the [M+H]^+^ form of M11 and M15 was 440.22 suggesting tri-oxidation metabolites. The product ion scan of *m/z* 440.22 led to the formation of fragment ions at *m/z* 293.19, 172.13, 148.04, 120.04 and 105.03. Among them, two unique fragment ions (293.19 and 172.13) increased by 32 amu and the other two fragment ions (148.04 and 120.04) increased by 16 amu showed that metabolism occurred in 4-butyl piperidine and pyridine-acryloyl moiety of Daporinad, respectively. The detailed fragmentation patterns of M11 and M15 are shown in Figure 7h.

M12

The *m/z* of the [M+H]^+^ form of M5 was 394.25 increased by 2 amu suggesting a saturation metabolite. The product ion scan of *m/z* 394.25 led to the formation of fragment ions at *m/z* 261.20, 140.14, 134.06 and 105.03. In the fragment pattern, a fragment ion increased by 2 amu showed that the metabolism of M12 was likely occurred in the fragment ion of *m/z* 132.04. The detailed fragmentation patterns of M12 are shown in Figure 7i.

M16 and M23

The *m/z* of the [M+H^]+^ form of M16 and M23 was 390.22, decreased by 2 amu suggesting desaturation metabolites. The product ion scan of *m/z* 390.22 led to the formation of fragment ions at *m/z* 259.18, 242.15, 138.13, 132.04, 105.03 and 104.05. In the fragment pattern of M16 and M23, three fragment ions (259.18, 242.15 and 138.13) decreased by 2 amu showed that the metabolism was occurred in 4-butyl piperidine of parent drugThe detailed fragmentation patterns of metabolites are shown in Figure 7j.

M18, M19 and M25

The *m/z* of the [M+H]^+^ form of M18 M19 and M25 was 406.21, suggesting that desaturation and oxidation took place based on the 14 amu increase from Daporinad. The product ion scan of *m/z* 406.21 led to the formation of fragment ions at *m/z* 259.18, 242.15, 148.04, 138.13, 120.04 and 105.03. In the fragment pattern of M18, M19 and M25, the three fragments (259.18, 242.15 and 138.13) with a decrease in *m/z* by 2 amu indicated the metabolism on 4-butyl piperidine of parent drug, and two fragments (148.04, 120.04) with an increase in *m/z* by 16 amu indicated the metabolism on the pyridine-acryloyl of parent drug. The detailed fragmentation patterns of M18, M18 and M25 are shown in Figure 7k.

M20

The *m/z* of the [M+H]^+^ form of M1 was 276.16. The product ion scan of *m/z* 276.16 led to the formation of fragment ions at *m/z* 170.12 and 105.03. These results indicated that M20 was formed by the hydrolysis of the amide bond of Daporinad followed by the conversion of the amine group to carboxylation. The detailed fragmentation patterns of M20 are shown in Figure 7l.

M21

The *m/z* of the [M+H]^+^ form of M21 is 262.18. The molecular ions are 130 amu less than Daporinad. The product ion scan of *m/z* 262.18 led to the formation of fragment ions at *m/z* 244.17 and 105.03. The *m/z* 244.17 fragment was not changed when compared with Daporinad; however, the *m/z* 262.18 was increased by 1 amu at *m/z* 261.2, suggesting that the amine group was possibly converted to a hydroxyl group. The detailed fragmentation patterns of M21 are shown in Figure 7m.

M22 and M24

The *m/z* of the [M+H]^+^ form of M22 and M24 was 408.23, suggesting mono-oxidation metabolites based on the 16 amu increase from Daporinad. The product ion scan of *m/z* 408.23 led to the formation of fragment ions at *m/z* 261.2, 244.17, 140.14, 148.04, 120.04 and 105.03. In the fragment pattern of M22 and M24, the two fragments (148.04, 120.04) increased by 16 amu indicated the metabolism to pyridine-acryloyl moiety of the parent drug. The detailed fragmentation patterns of M22 and M24 are shown in Figure 7n.

## 3. Materials and Methods

### 3.1. Reagents and Chemicals

Daporinad (99.94%) was purchased from MedChem Express (Monmouth Junction, NJ, USA). Verapamil, glutathione (GSH), uridine-50-diphosphoglucuronic acid triammonium salt (UDPGA) were purchased from Sigma-Aldrich (Yong-in, Gyeonggi, Korea). Dimethyl sulfoxide (DMSO), formic acid (FA) and methanol (MeOH) were obtained from Daejung reagents (Siheung, Gyeonggi, Korea). HPLC grade acetonitrile (ACN) and distilled water (DW) were from Samchun chemical (Pyeongtaek, Gyeonggi, Korea).

ICR mouse and human liver microsomes, β-nicotinamide adenine dinucleotide hydrate (NADPH) regenerating system solution A and B were from Corning (Tewksbury, MA, USA) and TiCl_3_ was bought from Kanto chemical (Tokyo, Japan). All other chemicals and reagents were analytical grade and purchased from commercial sources.

### 3.2. Preparation of Stock, Standard (STD) Working and Quality Control (QC) Working Solutions

A stock solution of Daporinad was prepared by dissolving the powder of Daporinad in DMSO to make concentration of 1 mg/mL and stored at −20 °C. A sub-stock solution was made by diluting stock solution with 10-fold. Sub-stock solution was serially diluted with DMSO to obtain final concentrations of 1.02, 3.05, 9.14, 27.4, 82.3, 247, 741 and 2220 ng/mL for STD working solutions and 15.0, 165 and 1820 ng/mL for QC working solutions. A stock solution of verapamil was also prepared by dissolving powder of compound in DMSO to make concentration of 1 mg/mL for internal standard (ISTD). The ISTD solution was made to prepare a final concentration of 100 ng/mL in ACN prior to sample preparation.

### 3.3. Sample Preparation for Method Qualification and PK Analysis

An aliquot of 4 µL of STD or QC working solutions were spiked into 20 µL of blank mouse plasma, and 4 µL of make-up DMSO was spiked into 20 µL of the mouse PK sample to make it the same matrix condition as the STD and QC sample. Then, 100 µL of ISTD solution was added to STD, QC and the study sample. Then, the samples were vortexed for about 1 min and centrifuged at 10,000× *g* rpm for 5 min. After centrifugation, 75 µL of supernatant was taken and mixed with 75 µL of DW. After that, the aliquot was transferred to an LC vial for the LC-qTOF-MS analysis.

### 3.4. Sample Preparation for In Vitro Met ID In Mouse and Human Liver Microsomes

First, a cofactor-compound mixture was prepared by mixing NADPH solution A and B, UDPGA, GSH and a working solution (2 mg/mL) of Daporinad. The above cofactor-compound mixture was pre-incubated at 37 °C for 5 min. 380 µL of the pre-incubated mixture was then transferred to 1.7 mL polypropylene tube, and 20 µL of 20 mg/mL mouse or human liver microsomes were applied. The microsomes and cofactor-compound mixture were incubated at 37 °C for 0 and 120 min and the reaction was stopped by adding 450 µL of ACN for protein precipitation. After centrifugation at 8000× *g* rpm for 10 min, 550 µL of supernatants was transferred to a fresh tube and evaporated to dryness under vacuum using rotary evaporator (Eyela CVE-3110 and UT-1000, Tokyo, Japan). Completely dried tubes were reconstituted with 110 µL of 30% ACN in DW with 0.1% FA. Reconstituted samples were centrifuged at 12,000× *g* rpm for 5 min, and its supernatant was transferred to an LC vial for the in vitro Met ID analysis.

### 3.5. Sample Preparation for In Vivo Met ID in Mouse PK Samples

For the in vivo Met ID analysis, plasma samples from 30 mg/kg IV PK groups were pooled according to the Hamilton pooling method, respectively [37]. 210 µL of the pooled plasma sample was transferred to a polypropylene tube and 800 µL of ACN was added for protein precipitation. Then, the above samples were centrifuged at 10,000× *g* rpm for 10 min and 900 µL of the supernatant was evaporated to dryness under vacuum in a rotary evaporator. Completely dried tubes were reconstituted with 110 µL of 30% ACN in DW with 0.1% FA. The reconstituted sample was vortexed and centrifuged at 12,000× *g* rpm for 5 min and the supernatant was transferred into a LC vial for the in vivo Met ID analysis.

### 3.6. Sample Preparation for N-Oxide Metabolites Identification in In Vitro and In Vivo Samples

To investigate the structural elucidation of N-oxide metabolites, 10 µL of TiCl_3_ solution was added to 90 µL of the pretreated in vitro and in vivo Met ID samples. Then, the mixtures were incubated at RT for 1 h. After that, the final mixed solution was transferred to a LC vial.

### 3.7. LC-qTOF-MS Condition

The LC-qTOF-MS assay consisted of a chromatographic pump system (Shimadzu CBM-20A/LC-20AD, Shimadzu Corporation, Columbia, MD, USA), an auto-sampler system (Eksigent CTC HTS PAL, LEAP Technologies, Carrboro, NC, USA), equipped with a mass spectrometer (TripleTOF^TM^ 5600, Sciex, Foster City, CA, USA). A Kinetex XB-C18 analytical column (2.1 × 50 mm, 2.6 μm; Phenomenex, Torrance, CA, USA) was used for bioanalytical sample quantification and Kinetex XB-C18 analytical column (2.1 × 100 mm, 2.6 μm; Phenomenex) was used for Met ID. C18 Security Guard Cartridge (4 × 2 mm; Phenomenex) was placed upstream of the analytical column. Mobile phase A (0.1% FA in DW) and mobile phase B (0.1% FA in ACN) were used following an optimized LC gradient. The LC gradient for quantification was optimized as follows: 3.0 min (0–0.5 min, 10% B; 0.5–0.95 min, 10–95% B; 0.95–1.4 min, 95% B; 1.4–1.5 min, 95–10% B; and 1.5–3.0 min, 10% B with a flow rate of 0.4 mL/min). The LC gradient for Met ID was also optimized as follows: 40 min (0–1 min, 10% B; 1–34 min, 10–35% B; 34–35 min, 35% B; 35–36 min, 35–95% B; 36–37.5 min, 95% B; 37.5–37.6 min, 95–10% B; 37.6–40 min, 10% B with a flow rate of 0.3 mL/min).

The curtain gas (CUR) was 33 L/min, the gas sources 1 and 2 were 50 psi, the ion spray voltage (ISVF) was set at 5500 V and the source temperature was 500 °C. For bioanalytical sample quantification, the high-resolution time of flight-mass spectrometer (TOF-MS) scan (Mass range: *m/z* 100–800, DP: 100 and CE: 10) and two product ion scans for Daporinad and verapamil using single reaction monitoring at high resolution option mode was used for the PK sample analysis. The product ion scans were performed by the following mass transitions *m/z* 392.2 to 105.2 (DP: 100 and CE: 40) for Daporinad and *m/z* 455.3 to 165.1 (DP: 125 and CE: 30) for verapamil, respectively.

For Met ID analysis, the high-resolution TOF full scan and eleven product ion scans were performed for *m/z* of Daporinad and its suspected metabolites. The following condition were used; TOF-MS scan (mass range: *m/z* 100–800, DP: 100 and CE: 10), product ion scans for Daporinad and its proposed metabolites (DP: 100 and CE: 40).

### 3.8. Method Qualification

The method qualification was performed with a ‘fit-for-purpose’ approach. The calibration curve consists of eight levels of STD and three levels of QC were used to evaluate the accuracy and precision for the qualification run. The intra-run accuracy and precision assay were evaluated using triplicates of QC samples. Dilution integrity assessments were performed to quantify samples whose concentrations in PK samples exceeded the upper limit of quantification (ULOQ). Three dilution factor (5, 10 and 30) sets consisting of triplicates were evaluated.

Preliminary stability assessments were conducted to demonstrate that the Daporinad is stable during various storage conditions and sample preparation process. Three levels of QC were used for the short-term, freeze–thaw and long-term stability assessments. The short-term stability assessment was carried out at RT for 4 h, and the long-term stability samples were kept frozen at −80 °C for 2 weeks. The freeze–thaw stability was evaluated over three cycles of freezing (−80 °C) and thawing (RT). The stability test results were evaluated by confirming that the mean accuracy (%) of the nominal value which should be within ±25% acceptance criteria.

### 3.9. Application for Animal Study

Male ICR mice (30 ± 3 g) were purchased from the Samtako biokorea co. (Pocheon, Gyeonggi, Korea) and housed in groups of 6~8 per cage and given standard rodent chow. The animals were fasted overnight with free access to water for at least 12 h before administration. Mice were distributed into three different dosing groups (n = 3 for each dosing group; 5, 10 and 30 mg/kg). The blood sampling time points were 2, 10, 30, 60, 120, 240 and 420 min after administration. The sampling blood samples were centrifuged at 12,000× *g* rpm for 5 min and the plasma was transferred to another tube and then stored at −20 °C until further analysis. All experiment performed on the mouse were approved by abiding the animal care protocol (no. CNU-01104) from Chungnam National University. The procedures were abided by the guidelines established by the Association for Assessment and Accreditation of Laboratory Animal Care International (AAALAC International).

### 3.10. Software

Data acquisition and LC-qTOF-MS operation was conducted using Analyst^®^ TF Version 1.6 (Sciex, Foster City, CA, USA). MultiQuant^®^ Version 2.1.1 (Sciex, Foster City, CA, USA) was used for the peak integration for Daporinad quantification. PeakView^®^ Version 2.2 (Sciex, Foster City, CA, USA) and MetabolitePilot™ Version 2.0.2 (Sciex, Foster City, CA, USA) were used for the structural elucidation of Daporinad metabolites. The descriptive statistics for the qualification studies were calculated with Excel 2015 (Microsoft). Pharmacokinetic parameters were calculated in a non-compartmental analysis using WinNonlin^®^ version 8.0.0 (Certara, Princeton, NJ, USA).

## 4. Conclusions

In this study, an LC-qTOF-MS assay was developed and qualified for the quantification of Daporinad in mouse plasma. The calibration curves were acceptable over the concentration range from 1.02 to 2200 ng/mL for Daporinad using the quadratic regression with a correlation coefficient ≥ 0.99. Daporinad was stable in mouse plasma under the several preliminary stability test conditions, such as short-term (4 h), freeze–thaw (three cycles), and long-term (2 weeks), and achieved the dilution integrity. This method was successfully applied to quantify the in vivo IV PK mouse plasma samples.

The PK results suggest that Daporinad has low to moderate clearance values depending on the 5 to 30 mg/kg administered dose range. It was interesting that in the dose range from 10 to 30 mg/kg, the C_max_ was dose-proportional, but the AUC appeared to increase supra proportions. There are several possibilities regarding this result and based on the clearance mechanism, we hypothesize that either metabolic enzymes or transporters related to the elimination pathway of Daporinad might play a role. This triggers in vitro and in vivo MetID studies, and as a result, twenty-five metabolites were newly identified under the current experimental conditions. The results suggest that ten different metabolic pathways were identified for Daporinad, and most of them were phase Ⅰ metabolic reactions, such as amide hydrolysis, oxidation and desaturation. Although many interesting metabolites were newly identified in this experiment, no significant difference from in vivo metabolites perspectives were observed from different dose levels conducted in this study (data not shown) and, therefore, it appeared that other mechanism such as saturation of transporters etc might likely play a role for this phenomenon of clearance change we observed from the in vivo mouse PK. Further experiments such as semi-mass balance study to understand the elimination pathway or in vitro transporter assays to identify the responsible transporters would be necessary.

In conclusion, we developed a sensitive, simple and reproducible LC-qTOF-MS assay to evaluate Daporinad in mouse PK samples, and also evaluated the in vitro and in vivo metabolite profiling of Daporinad with several novel metabolites. This research would warrant further experiments to better understand the in vivo clearance mechanism of Daporinad.

## Figures and Tables

**Figure 1 molecules-27-02011-f001:**
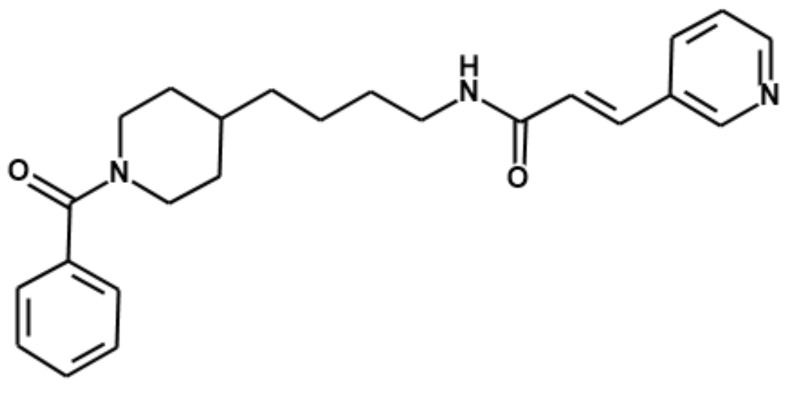
Structure of Daporinad.

**Figure 2 molecules-27-02011-f002:**
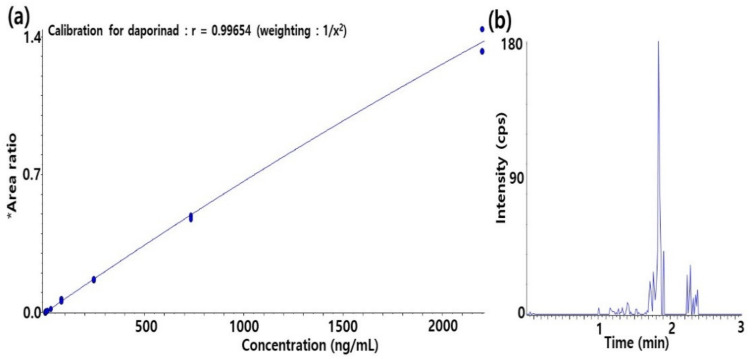
(**a**) Calibration curve of Daporinad in mouse plasma; (**b**) Chromatogram of the lower limit of quantification (LLOQ) level of Daporinad in mouse plasma. Y axis; * area ratio = [analyte peak area/internal standard peak area].

**Figure 3 molecules-27-02011-f003:**
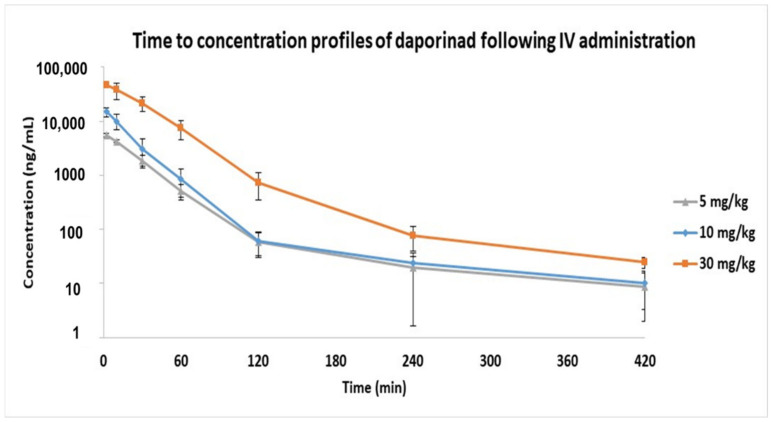
Time to concentration profiles of Daporinad following IV administration at doses of 5, 10 and 30 mg/kg (n = 3 for each dosing groups).

**Figure 4 molecules-27-02011-f004:**
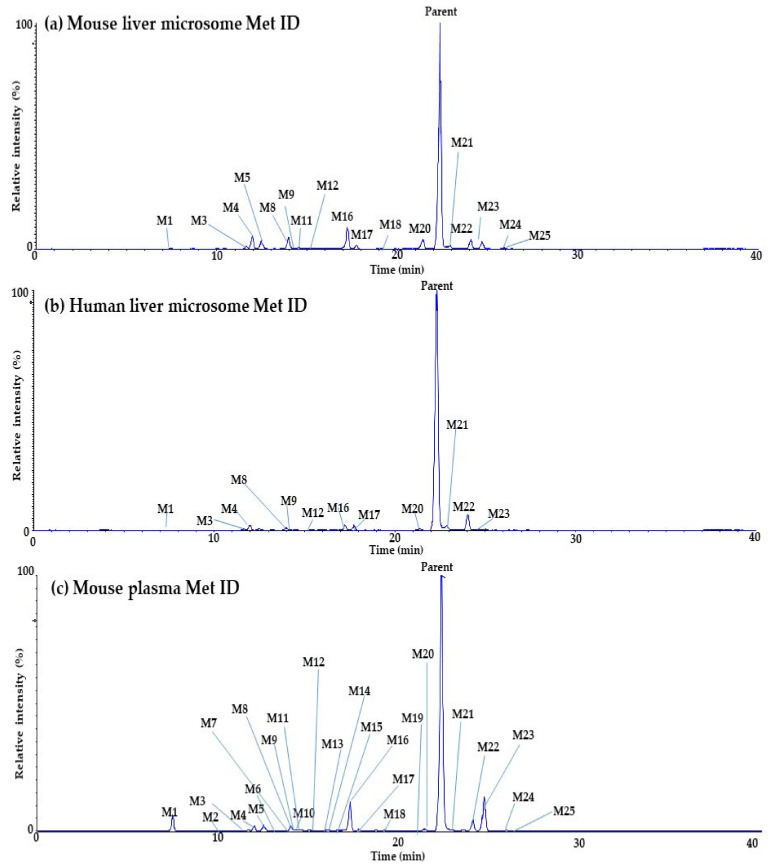
Extracted ion chromatograms of Daporinad and its metabolites in (**a**) mouse liver microsome, (**b**) human liver microsome, and (**c**) in vivo PK mouse plasma.

**Figure 5 molecules-27-02011-f005:**
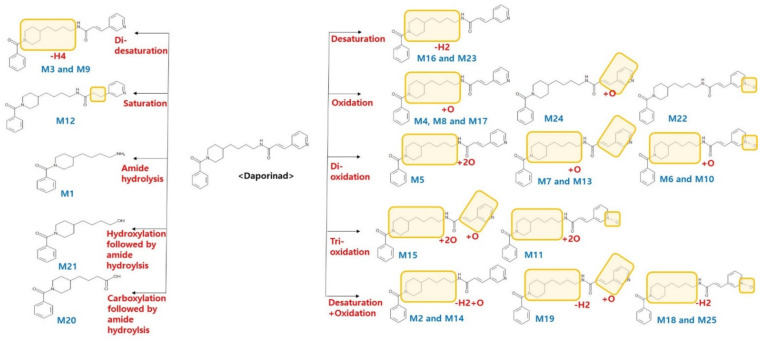
Metabolic pathways of Daporinad under in vitro and in vivo Met ID experiments.

**Figure 6 molecules-27-02011-f006:**
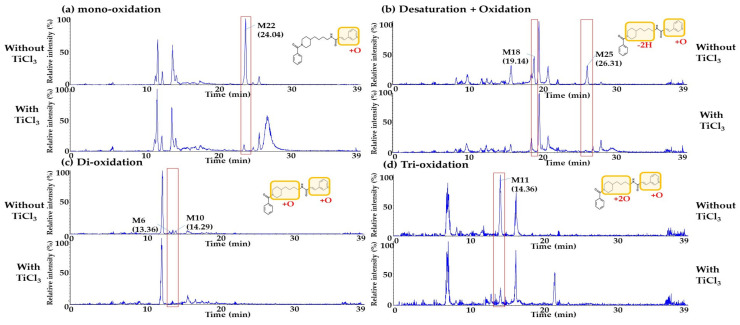
Product ion scan chromatograms of the oxidation metabolites with or without TiCl_3_ (**a**) N-oxide metabolite (M22: *m*/*z* 408.23); (**b**) N-oxide and desaturation metabolites (M18 and M25: *m*/*z* 406.21); (**c**) N-oxide and oxidation on the 4-butyl piperidine metabolites (M6 and M10: *m*/*z* 424.22) and (**d**) N-oxide and di-oxidation on the 4-butyl piperidine metabolite (M11: *m*/*z* 440.22).

**Figure 7 molecules-27-02011-f007:**
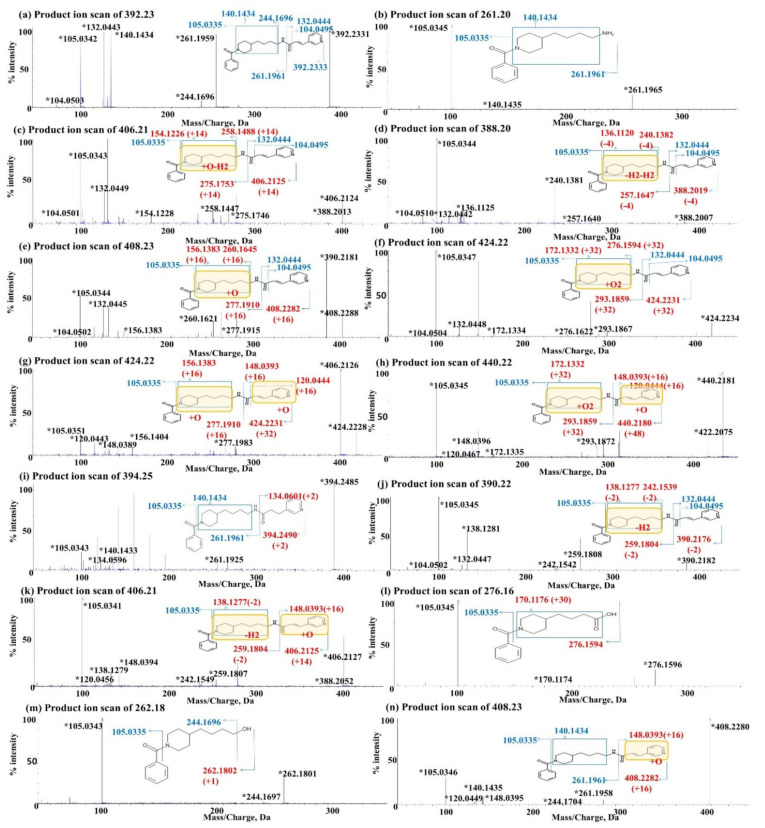
Product ion scan spectra of Daporinad and its metabolites. (**a**) Daporinad (*m/z* 392.23); (**b**) amide bond hydrolysis metabolite (M1: *m/z* 261.2); (**c**) oxidation and desaturation metabolites (M2 and M14: *m/z* 406.21), (**d**) di-desaturation metabolites (M3 and M9: *m/z* 388.2); (**e**) mono-oxidation metabolites (M4, M8 and M17: *m/z* 408.23); (**f**) di-oxidation metabolites (M5: *m/z* 424.22); (**g**) di-oxidation metabolites (M6, M7, M10 and M13: *m/z* 424.22); (**h**) tri-oxidation metabolites (M11 and M15: *m/z* 440.22); (**i**) saturation metabolite (M12: *m/z* 394.25); (**j**) desaturation metabolites (M16 and M23: *m/z* 390.22); (**k**) oxidation and desaturation metabolites (M18, M19 and M25: *m/z* 406.21); (**l**) amide bond hydrolysis followed by carboxylation metabolite (M20: *m/z* 276.15); (**m**) amide hydrolysis followed by hydroxylation metabolite (M21: *m/z* 262.18) and (**n**) mono-oxidation metabolites (M22 and M24: *m/z* 408.23). * Product ions used for structural elucidation.

**Table 1 molecules-27-02011-t001:** Quality control results and statistics from the intra/inter-run assays. For Daporinad in mouse plasma (n = 3 for each run assays).

Run	Statistics	Low QC (15.0 ng/mL)	Medium QC (165 ng/mL)	High QC (1820 ng/mL)
Intra-run 1	Mean accuracy (%)	96.9	107	104
	Precision (CV %)	10.3	6.00	10.1
Intra-run 2	Mean accuracy (%)	106	90.7	97.8
	Precision (CV %)	12.7	4.16	1.68
Intra-run 3	Mean accuracy (%)	106	109	102
	Precision (CV %)	10.7	4.64	7.72
Inter-run	Mean accuracy (%)	103	102	101
	Precision (CV %)	10.1	8.91	6.71

**Table 2 molecules-27-02011-t002:** The dilution integrity assessment of Daporinad in mouse plasma (n = 3 for each assessment).

Assessment	Dilution Factor	Statistics	Dilution QC
Dilution integrity	5-fold	Theoretical concentration (ng/mL)	6670
		Mean concentration (ng/mL)	7550
		Mean accuracy (%)	113
		Precision (%, CV)	3.12
	10-fold	Theoretical concentration (ng/mL)	20,000
		Mean concentration (ng/mL)	19,300
		Mean accuracy (%)	96.5
		Precision (%, CV)	2.20
	30-fold	Theoretical concentration (ng/mL)	6670
		Mean concentration (ng/mL)	6220
		Mean accuracy (%)	93.2
		Precision (%, CV)	4.45

**Table 3 molecules-27-02011-t003:** Preliminary stability results for Daporinad in mouse plasma (n = 3 for each assessment).

Stability Assessment	Statistics	Low QC (15.0 ng/mL)	Medium QC (165 ng/mL)	High QC (1820 ng/mL)
Short-term (4 h, RT)	Mean accuracy (%)	103	102	104
	Precision (CV %)	7.22	2.54	3.08
Long-term (14 days, −20 °C)	Mean accuracy (%)	111	84.5	91.4
	Precision (CV %)	3.71	11.7	4.17
Freeze–thaw (3 cycles, −80 °C)	Mean accuracy (%)	105	117	116
	Precision (CV %)	5.10	5.18	2.61

**Table 4 molecules-27-02011-t004:** PK parameters of Daporinad following IV administration at the doses of 5, 10 and 30 mg/kg (n = 3 for each dosing groups).

Dose (mg/kg)	T_1/2_ (min)	C_max_ (ng/mL)	AUC_last_ (min∙ng/mL)	Cl (mL/min/kg)	V_ss_ (mL/kg)
5	52.91 ± 7.30	5476.2 ± 426.73	171,690.41 ± 19,880.05	29.29 ± 3.70	897.06 ± 199.49
10	50.49 ± 4.88	14,872.07 ± 2653.47	357,205.55 ± 36,369.09	28.13 ± 2.78	646.57 ± 91.48
30	40.76 ± 0.95	48,059.1 ± 3920.65	1,763,266.5 ± 457,583.63	17.88 ± 5.11	535.76 ± 110.37

T_1/2_= half-life, C_max_ = maximum concentration, AUC_last_ = area under the curve by the last time point, Cl = clearance, V_ss_ = volume of distribution at the steady state.

**Table 5 molecules-27-02011-t005:** Characterization of Daporinad and its metabolites using the LC-qTOF-MS assay.

Symbol	Metabolites	*m/z*	Formula	Retention Time (min)	Error ppm	Mouse Liver Microsome	Human Liver Microsome	Mouse Plasma
M1	Amide hydrolysis	261.1961	C_16_H_24_N_2_O_4_	7.36	0.4	O	O	O
M2	Desaturation + Oxidation	406.2125	C_24_H_27_N_3_O_3_	10.07	0.2	-	-	O
M3	Di-desaturation	388.2020	C_24_H_31_N_3_O_2_	11.53	1.0	O	O	O
M4	Oxidation	408.2282	C_24_H_29_N_3_O_3_	11.99	0.0	O	O	O
M5	Di-oxidation	424.2231	C_24_H_29_N_3_O_4_	12.48	0.2	O	-	O
M6	Di-oxidation	424.2231	C_24_H_29_N_3_O_4_	13.46	1.9	-	-	O
M7	Di-oxidation	424.2231	C_24_H_29_N_3_O_4_	13.91	2.1	-	-	O
M8	Oxidation	408.2282	C_24_H_29_N_3_O_3_	14.00	0.2	O	O	O
M9	Di-desaturation	388.2020	C_24_H_31_N_3_O_2_	14.11	1.0	O	O	O
M10	Di-oxidation	424.2231	C_24_H_29_N_3_O_4_	14.29	0.2	-	-	O
M11	Tri-oxidation	440.2180	C_24_H_29_N_3_O_5_	14.41	0.7	O	-	O
M12	Saturation	394.2489	C_24_H_31_N_3_O_2_	15.37	1.0	-	-	O
M13	Di-oxidation	424.2231	C_24_H_29_N_3_O_4_	15.87	0.7	-	-	O
M14	Desaturation + Oxidation	406.2125	C_24_H_27_N_3_O_3_	16.04	1.0	-	-	O
M15	Tri-oxidation	440.2180	C_24_H_29_N_3_O_5_	16.54	1.4	-	-	O
M16	Desaturation	390.2176	C_24_H_27_N_3_O_2_	17.27	2.8	O	O	O
M17	Oxidation	408.2282	C_24_H_29_N_3_O_3_	17.76	2.0	O	O	O
M18	Desaturation + Oxidation	406.2125	C_24_H_27_N_3_O_3_	19.15	0.7	O	-	O
M19	Desaturation + Oxidation	406.2125	C_24_H_27_N_3_O_3_	21.02	0.7	-	-	O
M20	Amide hydrolysis followed by carboxylation	276.1594	C_16_H_21_NO_3_	21.25	1.8	O	O	O
Parent	Parent	392.2333	C_24_H_29_N_3_O_2_	22.31	2.3	O	O	O
M21	Amide hydrolysis followed by hydroxylation	262.1802	C_16_H_23_NO_2_	22.76	0.4	O	O	O
M22	Oxidation	408.2282	C_24_H_29_N_3_O_3_	24.04	1.2	O	O	O
M23	Desaturation	390.2176	C_24_H_27_N_3_O_2_	24.67	2.8	O	O	O
M24	Oxidation	408.2282	C_24_H_29_N_3_O_3_	25.90	0.7	O	-	O
M25	Desaturation + Oxidation	406.2125	C_24_H_27_N_3_O_3_	26.31	0.5	O	-	O

## Data Availability

The data presented in this study are available on request from the corresponding author.

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
