# Peer review of "Quantitative Analysis of Daporinad (FK866) and Its In Vitro and In Vivo Metabolite Identification Using Liquid Chromatography-Quadrupole-Time-of-Flight Mass Spectrometry"

_molecules, 2022, doi:10.3390/molecules27062011_

Round 1

Reviewer 1 Report

The present manuscript entitled "Quantitative Analysis of Daporinad (FK866) and Its In vitro and In vivo Metabolite Identification Using Liquid Chromatography-Quadrupole-Time-Of-Flight Mass Spectrometry" by Minjae Park, Byeong ill Lee, Jangmi Choi, Yuri Park, Seo-jin Park, Jeong-hyun Lim, Jiyu Lee, and Young G. Shin (molecules-1649950) describes the development of a simple and sensitive liquid chromatography-quadrupole time of flight mass spectrometry method to investigate the pharmacokinetic properties of daporinad in mouse plasma. Additionally, in vitro and in vivo metabolite identification was also checked.

The present article is written correctly and has a good structure; moreover, it has all the necessary parts. The article is interesting from an analytical point of view; therefore, it should interest the reader. The paper meets Molecules' requirements, and I recommend the article for publication in Molecules following the common editing stage. My current decision is a major revision. More specific comments and observations are presented below.

  1. The most important consideration concerns the validation of the analytical procedure. If a new method is described, it should be validated, which is currently missing in the article. Please add a section on the validation of the analytical procedure. Validation parameters can be mentioned in the abstract.
  2. Introduction. One paragraph about analytical techniques for testing this compound may be added.
  3. Figures. Giving values only at the beginning and end of the axis is not informative. Please add a few values between the beginning and the end.
  4. Figure 2a. Please, replace commas with dots.
  5. Has the interference been studied? What can be done in the event of strong interference effects? How would you deal with them? What types of interference effects could occur?
  6. Figure 3. The drawing border should be removed. The axes should be more clearly marked.
  7. Table 4. Explanations of the abbreviations used should be placed under the table. Please check that all used abbreviations are explained beforehand.
  8. Figure 5. The inscriptions are not clearly visible.
  9. Figure 6. You can enlarge the axis description along with the values. The chemical formulas are strangely stretched. The red frame is too thick.
  10. Figure 7. The chemical formulas are strangely stretched. The inscriptions are not clearly visible.
  11. Section 3.1. A very monotonous description. Tu much “were purchased”.
  12. References. Please adapt it to the requirements of the journal. Journal abbreviations are not used.

I hope that the comments presented will help improve the article.

Reviewer 2 Report

1. Introduction: the authors should discuss in detail the following publication dealing with the same topic

Qu, S.-D., Liu, G.-X.
Daporinad in vitro metabolite profiling via rat, dog, monkey and human liver microsomes by liquid chromatography/quadrupole-orbitrap mass spectrometry (2021) Rapid Communications in Mass Spectrometry, 35 (18), art. no. e9150

2. Linearity: please check the significant figures; is 1.02 a valid concentration level?

3. Tables 1, 2 and 3: please correct the significant figures.

4. Validation parameters are missing; please refer to the guidelines you followed for method validation.

5. Lines 180-285: can be presented as supplementary material.

6. Section 3.1: please provide purities where approapriate.

7. Line 340: Is there a specific reason for reconstitution in 110 μL and not in 100 μL?

8. Please provide the ethical permission number / code for animal experiments 

Round 2

Reviewer 1 Report

Dear Authors,

Thank you for your meticulous consideration of my comments. The paper has improved substantially and, to my opinion, is suitable for publication.

Reviewer 2 Report

The revised version is satisfactory and suitable for publication.